# Multivariate genomic architecture of cortical thickness and surface area at multiple levels of analysis

Andrew D. Grotzinger [1,2] ✉, Travis T. Mallard [3,4,5,8], Zhaowen Liu[3,4,5,8], Jakob Seidlitz [6,7], Tian Ge[3,4,5] & Jordan W. Smoller[3,4,5]

Recent work in imaging genetics suggests high levels of genetic overlap within cortical regions for cortical thickness (CT) and surface area (SA). We model this multivariate system of genetic relationships by applying Genomic Structural Equation Modeling (Genomic SEM) and parsimoniously define five genomic brain factors underlying both CT and SA along with a general factor capturing genetic overlap across all brain regions. We validate these factors by demonstrating the generalizability of the model to a semi-independent sample and show that the factors align with biologically and functionally relevant parcellations of the cortex. We apply Stratified Genomic SEM to identify specific categories of genes (e.g., neuronal cell types) that are disproportionately associated with pleiotropy across specific subclusters of brain regions, as indexed by the genomic factors. Finally, we examine genetic associations with psychiatric and cognitive correlates, finding that broad aspects of cognitive function are associated with a general factor for SA and that psychiatric associations are null. These analyses provide key insights into the multivariate genomic architecture of two critical features of the cerebral cortex.

The human cerebral cortex broadly refers to the brain's outer sheet of gray matter and is typically indexed using two central metrics: cortical thickness (CT) and surface area (SA). In practice, CT is operationalized as the distance between pial surfaces and white matter, and SA as geodesics along the gray-white matter boundary. These two measures are both key predictors of important life outcomes; for example, CT has been associated with a range of psychiatric disorders[1,2], and SA with a host of cognitive outcomes across the lifespan[3–6]. In the last decade, twin studies have shown that both metrics are highly heritable, while characterized by distinct genetic underpinnings[7,8]. Even more recently, genotyped samples with neuroimaging data have become large enough to employ genome-wide association studies (GWAS) as a means of identifying the specific genetic variants associated with these structural phenotypes. For example, the ENIGMA consortium examined bilateral averages of 34 cortical brain regions to identify 175 and 48 genetic loci associated with regional SA and CT, respectively[9]. Two additional findings include the observation, consistent with the family-based literature, that the different SA and CT brain regions were highly positively correlated within measures of CT and SA and negatively correlated across CT and SA[9]. These results point towards distinct, multivariate genetic architectures.

The current study utilizes large-scale, imaging genetics datasets to formally model the genetic overlap across brain regions within CT and SA using the Genomic Structural Equation Modeling (Genomic SEM)

[1]Institute for Behavioral Genetics, University of Colorado Boulder, Boulder, CO, USA. [2]Department of Psychology and Neuroscience, University of Colorado Boulder, Boulder, CO, USA. [3]Psychiatric and Neurodevelopmental Genetics Unit, Massachusetts General Hospital, Boston, MA, USA. [4]Center for Precision Psychiatry, Department of Psychiatry, Massachusetts General Hospital, Boston, MA, USA. [5]Stanley Center for Psychiatric Research, Broad Institute, Cambridge, MA, USA. [6]Department of Psychiatry, University of Pennsylvania, Philadelphia, PA, USA. [7]Department of Child and Adolescent Psychiatry and Behavioral Science, The Children's Hospital of Philadelphia, Philadelphia, PA, USA. [8]These authors contributed equally: Travis T. Mallard, Zhaowen Liu. ✉e-mail: Andrew.Grotzinger@colorado.edu

framework[10]. We began by performing exploratory and confirmatory factor analyses of the genetic correlations estimated from the ENIGMA CT and SA summary statistics. We then replicate this factor structure in a semi-independent sample from UK Biobank (UKB), showing that the multivariate structure identified in ENIGMA fits the data well for both the left and right hemispheres in UKB. Having established the portability of this factor structure, we characterize the genomic factors at three levels of analysis. First, we examined the boundaries that define these genomic factors by examining relationships with molecular, cellular, and functional topographical maps. Second, we applied Stratified Genomic SEM[11] to identify classes of genes that are enriched at the level of the structural imaging factors. Finally, we examined associations between the brain factors and both general and domain-specific facets of cognitive function and psychiatric disorders. Collectively, our multivariate analyses of structural imaging phenotypes provide key insights into the biological, functional, and clinical relevance of varying levels of structural brain organization.

## Results

### Genomic factor analysis

Our primary analyses utilize the ENIGMA GWAS summary statistics for $N \approx 33,992$ participants across the lifespan (age range: 3–91). These reflect 34 bilateral averages of regional CT and SA[9] defined using the Desikan–Killiany atlas segmentations[12]. We note that GWAS summary statistics were not corrected for a global structural metric (e.g., total SA or mean CT). Instead, global (co)variation was accounted for psychometrically by modeling a latent, general factor in the context of the bifactor model described below. This analytic pipeline has the advantage of avoiding bias due to adjusting for a heritable trait (i.e., global metrics for SA and CT)[13]. As expected, LD-score regression revealed high levels of genetic overlap across brain regions within each metric (Fig. 1), and individual regions all displayed highly significant levels of SNP-based heritability for both CT (average $h^2_{SNP} = 17.1\%$; range: 8.0–25.2%; $p \leq 9.10\text{E-}7$) and SA (average $h^2_{SNP} = 23.8\%$; range: 12.0–31.7%; $p \leq 5.47\text{E-}14$). We went on to model SA and CT separately given a moderate, negative genetic correlation across their global metrics ($r_g = -0.32$, $SE = 0.05$), previously described unique genetic underpinnings[9], and extant hypotheses about divergent developmental pathways across CT and SA[14].

We applied three tests (Kaiser[15], acceleration factor, and optimal coordinates[16]) to determine the optimal number of genomic factors that could be used to parsimoniously describe the data. We went on to fit exploratory factor analyses (EFAs) using the promax (i.e., correlated

factor) rotation based on these three tests. These EFA results were used to inform fitting confirmatory factor analytic models (CFAs) within Genomic SEM. More specifically, individual brain regions were assigned to a factor when their standardized loading was >0.5, or if the brain region did not achieve a loading of 0.5 for any factor, assigning the region to the factor with the largest standardized loading (additional details provided in "Methods"; see Fig. 1 for example path diagram). The CFAs were evaluated using standard metrics of model fit[10] (i.e., comparative fit index [CFI][17]; standardized root-mean-squared residual (SRMR); Akaike Information Criteria [AIC][18]). For all models, including expanded models that incorporate psychiatric and cognitive correlates, residual covariances were iteratively added to the model where indicated. This was done by obtaining the residual covariance matrix—calculated as the difference between the model-implied genetic covariance and observed genetic covariance matrix—and adding the residual covariances one at a time until they no longer reached a significance threshold of $p < 0.01$. This procedure resulted in adding eight and seven residual covariances for CT and SA, respectively.

The common factor model fit, implying a single factor on which all brain regions loaded, was acceptable for CT (AIC = 83732.6, CFI = 0.907, SRMR = 0.093) and did not fit the data well for SA (AIC = 179949, CFI = 0.826, SRMR = 0.076). A correlated factors model with five genomic factors defined by subsets of brain regions fit the data great and provided a better fit for both CT (AIC = 42220.0, CFI = 0.953, SRMR = 0.065) and SA (AIC = 60940.9, CFI = 0.941, SRMR = 0.057). Finally, a bifactor model fit the data best for CT (AIC = 33065.3; CFI = 0.964; SRMR = 0.063) and SA (AIC = 46120.3; CFI = 0.956; SRMR = 0.048). The bifactor model consisted of a general factor defined by all 34 brain regions along with five residual, uncorrelated factors capturing brain regions that covary above and beyond the global (general) structure (Supplementary Data 1–3 for full CFA and EFA model output; Supplementary Data 4 for model fit). We recognize that bifactor models are generally guaranteed to fit the data better, regardless of the data-generating process in the population[19]. At the same time, we consider the bifactor model informative as brain regions are known to globally covary, and because it provides a psychometrically informed comparison point to previous results produced using GWAS summary statistics that controlled for total SA and average CT. Thus, all analyses presented in the main text consider the bifactor model, while full results for the correlated factors model are presented in the Online Supplement. The average proportion of genetic variation in the individual brain regions explained by the general factor was 52.1% for CT

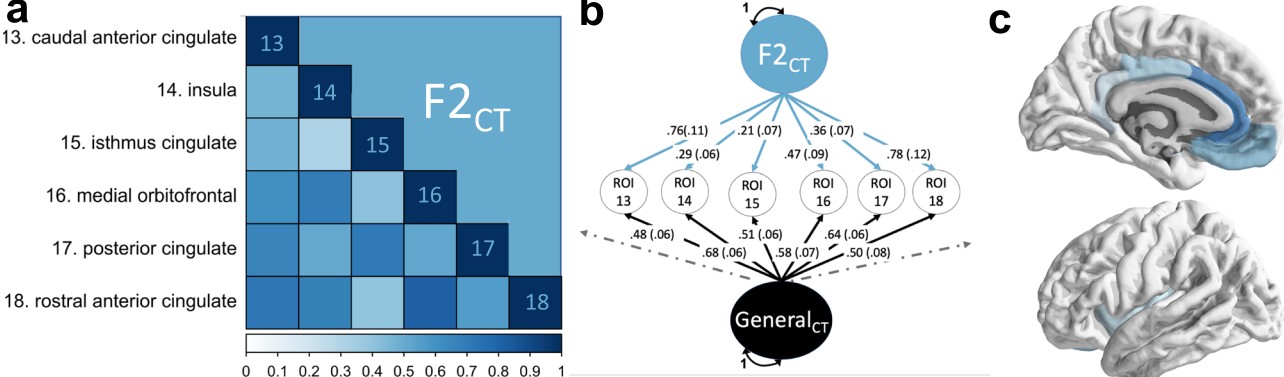

**Fig. 1 | Schematic of Genomic SEM. a** Truncated heatmap of genetic correlations for six regions of interest (ROIs) that all load onto the same genomic factor of cortical thickness (F2CT). These six brain regions were identified as loading on the same factor using exploratory factor analysis (EFA). **b** Path diagram of the confirmatory factor model results produced by Genomic SEM using the genetic correlation matrix in panel (**a**) as input. All parameter estimates are standardized with respect to SNP-based genetic variances. The genetic components of each brain region, the common factor defined by these genetic components, and the residual genetic variance for each brain region are represented as circles to reflect the fact that these are latent (i.e., not directly observed) variables. **c** Map of the factor standardized loadings for the six cortical regions on the cortex. Color coding is the same as in panel (**a**).

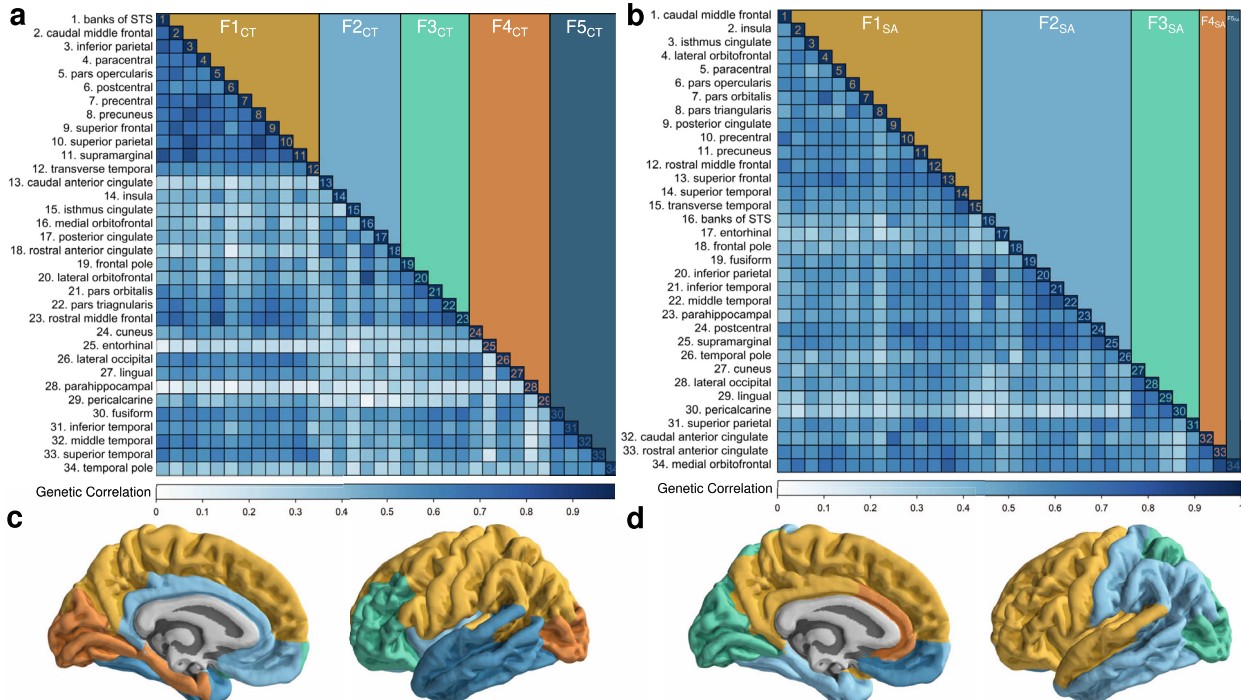

**Fig. 2 | Genomic factor analysis of the cerebral cortex. a, b** Heatmaps illustrating the genetic relationships between cortical regions for **a** CT and **b** SA, as estimated with Genomic SEM. Genetic correlations are reported in the lower triangle while factor membership is illustrated in the upper triangle. Cortical regions are ordered with respect to the factor model results. **c, d** Categorical brain maps depicting the brain regions that correspond to each genomic cortical factor for **c** CT and **d** SA. Color coding is consistent throughout the figure.

and 56.7% for SA. In addition, 24 of the CT factor loadings and 23 SA loadings on the five residual factors were significant at a Bonferroni-corrected threshold for 34 brain regions. In sum, model fitting results and significant residual covariation indexed by the factor loadings suggest that pervasive genetic overlap across regional measures of CT and SA are not merely reflective of a single dimension of macroscale organization.

As would be expected, physically proximal brain regions clustered within the five residual factors. For CT, these factors can be approximately described as representing somatomotor frontoparietal ($F1_{CT}$), mesolimbic ($F2_{CT}$), prefrontal ($F3_{CT}$), occipital ($F4_{CT}$), and temporal ($F5_{CT}$) brain regions. For SA, the first two factors were more diffuse, representing a combination of frontal dorsotemporal ($F1_{SA}$) and parietal ventrotemporal ($F2_{SA}$), followed by a third factor defined by occipitoparietal regions ($F3_{SA}$), and finally two highly specific factors, one reflecting the anterior cingulate regions ($F4_{SA}$), and a fifth factor defined solely by the medial orbitofrontal region ($F5_{SA}$). We note that these characterizations do not consistently describe every brain region that loads on the factor and refer the reader to Fig. 2 for a full list of the regions that load on each factor (also see Supplementary Figs. 1 and 2 for a full path diagram for CT and SA, respectively). In addition, these factors are further annotated in the context of established cytoarchitectonic and functional classifications presented below.

The replicability of these findings was tested using a semi-independent, genetically informative imaging dataset from UKB ($N = 26,739$; see "Methods" for details on sample overlap). The UKB sample also offered the opportunity to examine the portability of the identified factor structure for a more age-homogenous cohort (age range: 39–73). We found that the bifactor models fit the data well for both the left and right hemispheres in UKB (Supplementary Data 4 for model fit; Supplementary Data 5–8 for full model outputs; Supplementary Figs. 3–7 for heatmaps), indicating that the identified factor structure is relatively stable and captures a robust, bilaterally symmetrical pattern of genetic covariation.

## Topographical annotation of cortical genomic factors

We next examined whether the spatial organization of the CT and SA factors reflected a biologically and functionally relevant partitioning of the cortex. To test these hypotheses, we utilized spin-based methods to compare the factor analytic parcellation to several canonical and meta-analytic maps from the neuroimaging literature (see "Methods"). First, we asked whether there was a statistically nonrandom overlap in the assignment of regions of interest to the five residual CT and SA genomic factors. We find that the spatial organization of both metrics did significantly overlap with one another ($P_{spin} < 1e-4$), suggesting that the genomic factors reflect meaningful boundaries of cortical (co) variation that are partially consistent across these two morphological indices. Moreover, these genomic parcellations appear to capture biological differences of the cortical sheet, as comparisons to a digitized parcellation of von Economo and Koskinas's[20] cytoarchitectonic mapping of the cortex (Fig. 3a) also revealed significant overlap (CT $P_{spin} = 1.00e-4$, SA $P_{spin} = 3.82E-2$, Fig. 3b, c).

Consistent with the notion that these genomic factors differentiate broad areas of the cortex by their biological underpinnings, further topographical annotation identified myriad aspects of intracortical microstructure, laminar differentiation, cellular/neuronal density, neurotransmitter receptor density, and cell-type-specific transcriptional signatures that significantly varied as a function of the CT and SA factors (Fig. 3d, Supplementary Figs. 7–8, Supplementary Data 9). Generally, results suggest that the SA factors better captured interregional variation of these biological features relative to the CT factors (Fig. 3d). While similar effects were seen for some neurobiological features (e.g., $CB_1$, $D_1$, and μ-opioid receptor densities), CT- and SA-specific effects were also observed, such as In1 inhibitory neuron signatures that were more associated with CT and somatostatin interneuron transcriptional signatures more associated with SA. This indicates that the factor structures identified within CT and SA partially index unique biological signatures.

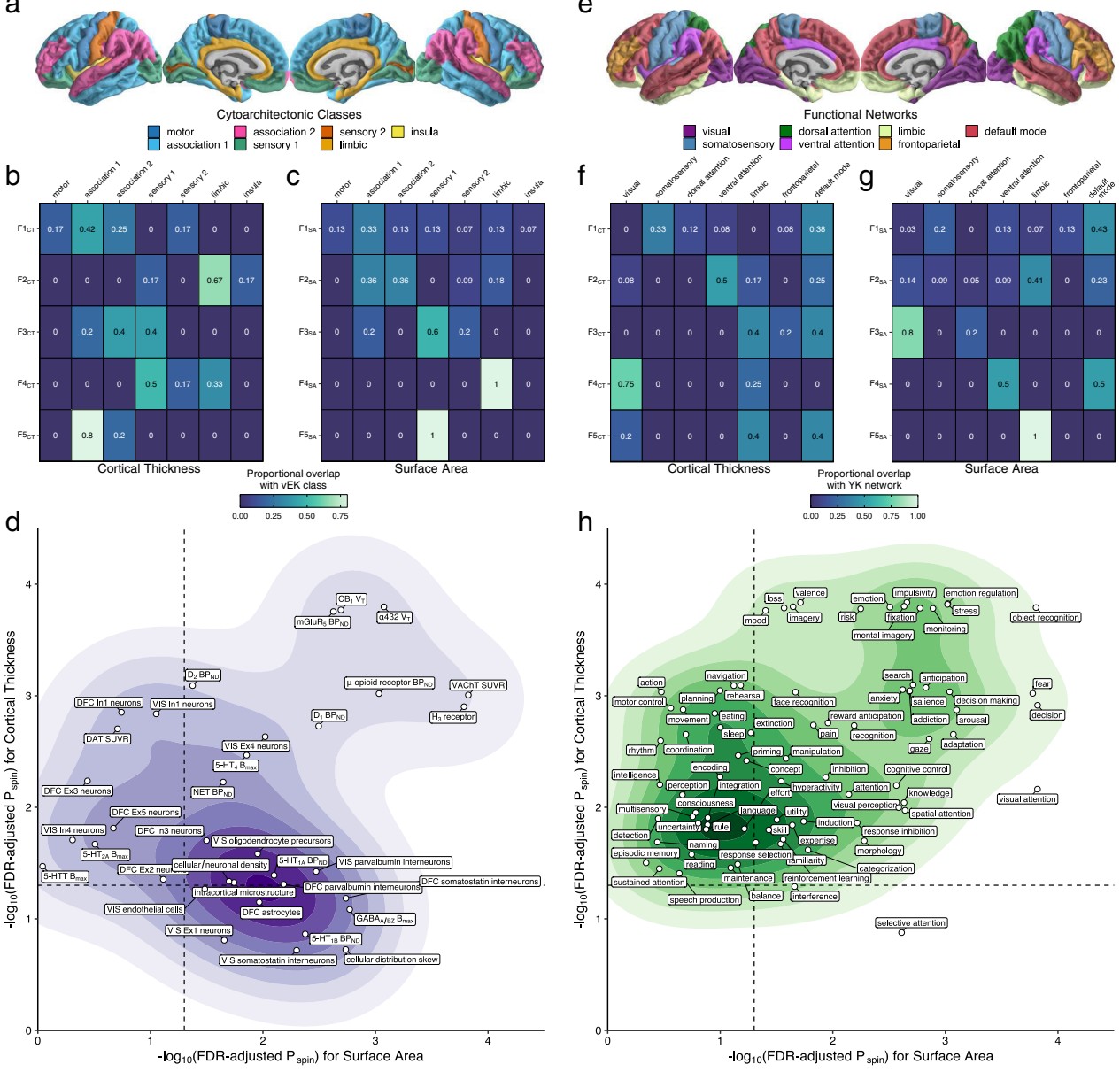

**Fig. 3 | Topographical annotation of genomic cortical thickness and surface area factors. a** Map of cytoarchitectonic classes defined by von Economo and Koskinas (vEK)[15]. **b, c** Marginal table of proportional overlap between **b** cortical thickness (CT) and **c** surface area (SA) and the vEK cytoarchitectonic classes. **d** Two-dimensional density plot of significant overlap between cortical factors and biologically derived features of the cortex (from the BigBrain project, Allen Human Brain Atlas, and positron emission tomography meta-analysis; "Methods"). Two-sided *p*-values were calculated using the spin-based test. The dashed line denotes statistical significance on a −log₁₀ scale after correcting for multiple comparisons

using the false discovery rate (FDR). **e** Map of canonical resting-state functional connectivity networks defined by Yeo and Krienen (YK)[16]. **f, g** Marginal table of proportional overlap between **f** CT and **g** SA and the YK networks. **h** Two-dimensional density plot of significant overlap between cortical factors and patterns of functional activation associated with psychological processes (from Neurosynth; "Methods"). As in panel (**d**), two-sided *p*-values were calculated using the spin-based test and the dashed line denotes statistical significance on a −log₁₀ scale after correcting for multiple comparisons using FDR.

Given these findings of molecular and cellular differences, we next asked how the genomic parcellations were organized relative to the functional topography of the cortical sheet. We first compared the spatial organization of the genomic factors to that of the seven canonical functional networks defined by Yeo and Krienen[21] (Fig. 3e) and found a significant overlap between these maps (CT $P_{spin}$ = 1.20E-3, SA $P_{spin}$ = 7.70E-3, Fig. 3f, g). To obtain finer-grained insights into how the genomic factors might relate to aspects of brain function, we then compared our factor analytic maps to meta-analytic maps of functional activation for 123 cognitive processes from Neurosynth ("Methods").

As observed with the biologically focused analyses above, we found that many patterns of functional activation associated with cognitive processes, emotion regulation, reward learning, decision-making, and visual processing were significantly different across the genomic factors for both CT and SA (Fig. 3h, Supplementary Figs. 9–10, Supplementary Data 9). However, in contrast to the more biologically derived maps, these results revealed that CT factors better captured inter-regional variation of psychological and cognitive features relative to the SA factors, with specific effects for perception, locomotion, and language.

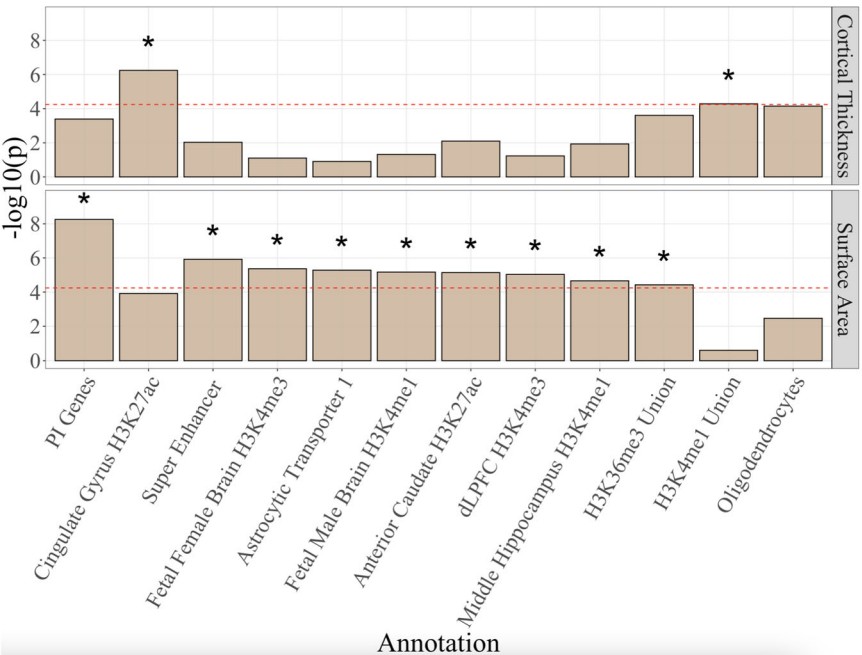

**Fig. 4 | Functional enrichment for general factors.** The figure depicts the 11 annotations that were Bonferroni significant for the general factor from the bifactor model for either surface area or cortical thickness, along with the oligodendrocytes annotation that was just below the Bonferroni threshold for CT ($p = 7.11E-5$). Both panels depict the $-\log10(p)$ values on the multivariate enrichment estimate from Stratified Genomic SEM. $p$-values are one-tailed and were calculated using the ratio of the enrichment estimate over its standard error. We correct for multiple testing for Stratified Genomic SEM results by employing a strict Bonferroni correction for the number of annotations (152) and factors (6) analyzed (i.e., $p < 5.48E-5$). Bars significant at a Bonferroni-corrected threshold (shown as a red dashed line) are depicted with a "*". Bars are ordered with respect to the level of significance across both CT and SA.

## Stratified Genomic SEM

Stratified Genomic SEM[11] is a multivariate analogue of partitioned heritability analyses[22] that can be used to determine whether a given functional annotation is enriched for any model parameter within Genomic SEM. Functional annotation is used here to denote a set of genetic variants that are grouped according to shared characteristics. For the current analyses, we specifically utilize 168 functional annotations reflecting genes expressed in different brain regions, evolutionarily conserved regions, tissue types, histone marks, neuronal cell types, and protein-truncating-variant-intolerant (PI) genes (see "Methods" for further details). A functional annotation is deemed enriched if the proportion of genetic variance explained by the annotation is greater than the proportional size of the annotation. Multivariate functional analyses then allow for distilling both highly polygenic and pleiotropic signals into their biologically meaningful, constituent parts. To this end, we specifically examined the enrichment of the genomic factor variances.

After removing 16 annotations that produced highly non-positive definite, stratified genetic covariance matrices (indicating unstable estimates), we examined a total of 152 annotations. Employing a Bonferroni correction of $p < 5.48E-5$ (i.e., $0.05/$ [$152_{annotations} \times 6_{Genomic\ Factors}$]), we identified two significant annotations for the general CT factor: H3K4me1 union and the cingulate gyrus H3K27ac histone mark. We highlight that the oligodendrocytes annotation was also just below the significant threshold for CT ($p = 7.11E-5$; Supplementary Data 10). There were nine significant annotations for the general SA factor, including the PI Genes, Super Enhancer, H3K4me3 histone mark in the fetal female brain, and astrocytic transporters annotations (Supplementary Data 11). Five histone mark annotations were also significant for the fifth SA factor defined solely by the medial orbitofrontal region (Supplementary Fig. 11). In line with the modest inverse correlation across these two metrics and unique topographical signatures described directly above, the patterns of enrichment were markedly different across the top functional annotations for SA and CT (Fig. 4).

## Genetic overlap with cognitive function

We examined the genetic overlap between the brain-based factors and a $g$-factor estimated from UKB GWAS summary statistics for seven cognitive traits: trail-making tests-B, tower rearranging, verbal numerical reasoning (VNR), symbol digit substitution, memory pairs-matching test, matrix pattern recognition, and reaction time (RT; Table 1). The genetic overlap across these seven cognitive indicators was modeled using the same common factor model for genetic $g$ identified by de la Fuente and colleagues[23]. Genetic correlations were estimated across $g$ and the brain-based factors for the separate CT and SA bifactor models. We first sought to test whether the degree of genetic overlap across $g$ and brain morphology also varied by factor. For both CT and SA, we found robust evidence that the genetic correlations between $g$ and the brain factors could not be constrained to be equal ($p < 0.05$ for both $\chi^2$ difference tests, "Methods"). This suggests that there is significant variation across the cortical sheet in the degree of genetic overlap between intelligence and brain structure.

We next turned our attention to both global and localized patterns of genetic overlap between the brain factors and $g$, allowing for residual covariances between individual brain regions and cognitive tests where indicated ("Methods"). As each family of tests for CT and SA with $g$ consisted of 204 possible associations (i.e., 34 brain regions × 7 cognitive tests), we employed a Bonferroni correction of $0.05/204$ ($p < 2.10E-4$). There were no significant factor correlations for CT in the bifactor model after correcting for multiple comparisons, though we identified a nominally significant residual genetic correlation across VNR and the precentral region (partial $r_g = 0.38$, $p = 2.50E-4$). Bifactor model results revealed a significant genetic correlation between the general SA and $g$-factor ($r_g = 0.24$, $SE = 0.04$, $p = 2.34E-11$). None of the correlations between $g$ and the five residual factors or individual brain regions were significant. These results indicate that the genetic overlap

**Table 1 | Summary of external traits**

| Cognitive traits | | |
| --- | --- | --- |
| **Trait** | **Data source** | **Total sample size** |
| Reaction time[23] | UKB | 330,024 |
| Matrix pattern recognition[23] | UKB | 11,356 |
| Verbal numerical reasoning[23] | UKB | 171,304 |
| Symbol digit substitution[23] | UKB | 87,741 |
| Memory pairs-matching test[23] | UKB | 331,679 |
| Tower rearranging[23] | UKB | 11,263 |
| Trail-making test-B[23] | UKB | 78,547 |

| Psychiatric traits | | | | |
| --- | --- | --- | --- | --- |
| **Trait** | **Data source** | **Cases** | **Controls** | **Population prevalence** |
| Alcohol use disorder[33] | PGC | 8485 | 20,272 | 15.90 |
| Anorexia nervosa[24] | PGC | 16,992 | 55,525 | 0.90 |
| Anxiety disorders[32] | UKB + ANGST + iPSYCH | 31,977 | 82,114 | 20.00 |
| Attention-deficit hyperactivity disorder[30] | PGC | 19,099 | 34,194 | 5.00 |
| Autism spectrum disorder[29] | PGC | 18,381 | 27,969 | 1.20 |
| Bipolar disorder[28] | PGC | 41,917 | 371,549 | 2.00 |
| Major depressive disorder | PGC + UKB | 170,756 | 329,443 | 15.00 |
| Obsessive-compulsive disorder[25] | PGC | 2688 | 7037 | 2.50 |
| Post-traumatic stress disorder[31] | PGC | 2424 | 7113 | 30.00 |
| Schizophrenia[27] | PGC | 53,386 | 77,258 | 1.00 |
| Tourette's syndrome[26] | PGC | 4819 | 9488 | 0.80 |

The top half of the table reports information for the cognitive phenotypes while the bottom half reports information for the 11 psychiatric traits. Sample sizes reported are for the European ancestry-only subsets of the summary statistics. We note that for many of the psychiatric summary statistics, an SNP-specific sum of effective sample sizes was provided and that these effective sample sizes were used for relevant calculations within Genomic SEM. The population prevalence column lists the prevalence used for liability scale conversion. The broad depression phenotype was specifically used from the UKB Major Depression GWAS. The lifetime anxiety disorder phenotype was used for the UKB Anxiety GWAS. The PGC ALCH GWAS used the unrelated subsample from the overall genotyped European sample. We note that the sample sizes listed are the raw totals and do not reflect effective sample sizes.

between SA and cognitive function can be conceptualized as operating through largely general pathways shared across brain regions and different classes of cognitive tasks. Supplementary Data 12 also compares genetic correlations obtained for the general factors to the estimates using the global averages across all 34 brain regions for CT and SA. As would be expected, the direction of effects across the global averages and general factors was consistent, while point estimates for the general factors estimated in Genomic SEM were slightly larger. This slight increase in estimates likely reflects improvement in genetic signal when psychometrically extracting what is shared across the 34 regions via the general factor, as opposed to taking the global average.

## Genetic overlap with psychiatric traits

To examine the multivariate system of genetic relationship with psychiatric traits, we employed the same four-factor correlated factors model of 11 major disorders identified in ref. [11]. This model consists of a Compulsive disorders factor defined by anorexia nervosa[24], obsessive-compulsive disorder (OCD)[25], and Tourette's syndrome[26], a Psychotic disorders factor defined by schizophrenia[27] and bipolar disorder[28], a Neurodevelopmental disorders factor defined by autism spectrum disorder (ASD)[29], attention-deficit hyperactivity disorder (ADHD)[30], and post-traumatic stress disorder (PTSD)[31], and an Internalizing disorders factor defined by major depressive disorder (MDD), anxiety disorders[32], and PTSD[31] (Table 1; Supplementary Data 13 for factor loadings from psychiatric measurement model). In addition, alcohol use disorder[33] loaded on the Psychotic, Neurodevelopmental, and Internalizing factors. As with the $g$-factor analyses, we tested whether the degree of genetic overlap between psychopathology and brain morphology varied across the cortex. For CT, the constrained model with invariant genetic correlations per psychiatric factor would not converge, suggesting it is not an appropriate model. Similarly, for SA we found robust evidence that the genetic correlations between the psychiatric factors and the brain factors could not be constrained to be

equal ($p < 0.05$, see "Methods"), indicating that there is significant variation across the cortex in the degree of genetic overlap with the psychiatric factors.

We then estimated genetic correlations between all psychiatric and brain factors and added residual covariances between individual disorders and brain regions when indicated. We employed a Bonferroni-corrected significance threshold of $p < 1.34\text{E-}4$ (i.e., 34 brain regions × 11 disorders = 0.05/374). There were no significant genetic correlations across any of the CT or psychiatric factors (Supplementary Data 14). The Neurodevelopmental factor displayed the strongest genetic correlation with the general SA factor in the bifactor model ($r_g = -0.17$, $SE = 0.05$, $p = 1.19\text{E-}3$), though this was not significant at a Bonferroni-corrected threshold. Four nominally significant residual relationships were also identified between ASD and specific brain regions, the strongest of which was with the rostral anterior cingulate region (partial $r_g = 0.184$, $SE = 0.051$, $p = 3.23\text{E-}4$). In summary, the associations with psychiatric traits were largely null at the level of both genomic factors and individual brain regions.

## Discussion

The current study examined the multivariate genomic architecture of CT and SA at various levels of analysis using two of the largest available imaging genomic datasets. At the genome-wide level, we find that ENIGMA summary statistics for 34 physically proximal brain regions can be grouped across a general factor and five residual factors. In addition, we observe that this factor structure fits the data well for the left and right hemispheres in UK Biobank, indicating a bilaterally symmetrical structure. We also find that these five genomic factors explain significant genetic variation in the individual brain regions even when pulling out shared global variation via the general factor. It is of note that the multivariate architectures with respect to the make-up of the genomic factors were distinct across CT and SA. This is in line with prevailing developmental accounts of these structures, including

the radial unit hypothesis that argues for distinct developmental trajectories categorized by neurogenetic division of neural progenitor cells for CT, as compared to the propagation of these cells for SA[14]. Indeed, previous studies have shown that CT and SA are characterized by differing developmental trajectories[34,35], and the family-based literature corroborates the finding that genetic variation in these two metrics is distinct [7,8].

To understand these genomic parcellations of the cortex, we used recently developed surface-based methods to place our findings in the broader context of the neuroimaging literature. This topographical annotation allowed us to evaluate the spatial organization of our CT and SA factors relative to two canonical parcellations of the cortex: the cytoarchitectonic classes defined by von Economo and Koskinas[20] and the functional networks derived by Yeo and Krienen[21]. For both canonical parcellations, we found significant patterns of overlap between classifications of cortical regions, suggesting the organization of the CT and SA factors reflected a biologically and functionally relevant partitioning of the cortex. Further annotation revealed relationships across the identified genomic factor structures and myriad quantitative features of the cortex, such as neurotransmitter receptor densities, cell-type transcriptional signatures, and patterns of functional activation measured by fMRI.

These results provide an anatomical context for our factor analytic findings and lend insight into what underlies the differences between CT and SA factors. For example, while the SA factors were more broadly related to biologically derived features of the cortex (e.g., neuronal density, neurotransmitter receptor densities), the CT factors were more strongly related to functional activation underlying a variety of psychological processes. Interestingly, the spatial organization of both CT and SA factors captured functional boundaries of the cortex underlying emotion regulation, reward learning, decision-making, and visual processing. At the functional genomic level, we find that astrocytic transporters and (consistent with their generalized biological roles) the PI genes and Super Enhancer annotations are broadly relevant for SA, with significant enrichment identified for the general factor. Notably, we do not observe enrichment for these annotations for CT, indicating that these classes of genes are specifically relevant to SA.

The current analyses reflect a well-powered, psychometrically informed approach to examining genetic overlap across CT, SA, and a broad range of psychiatric and cognitive outcomes. Consistent with prior work[3-6], we find that the genetic underpinnings for a general factor of SA are associated with a diverse set of cognitive functions, as indexed by a genetic $g$-factor. Conversely, we observe no significant genetic relationships across CT or SA and the four psychiatric factors. These null results are in line with recent findings indicating that the majority of associations across structural metrics and human complex traits are much smaller than initially thought and that the bulk of prior studies has been underpowered[36]. At the same time, large-scale phenotypic meta-analyses indicate widespread associations across structural metrics and various psychiatric disorders[37]. These findings may reflect associations that operate through largely environmental pathways or the current analyses may not be sufficiently powered to detect genetic effects for psychiatric disorders. It is also possible that psychiatric-structural associations are specific to different parcellations of the cortex or to clinically ascertained samples. In line with this latter account, associations between CT and various psychiatric disorders have been shown to reflect responses to treatment [38,39].

Prior work using dimension reduction techniques applied hierarchical clustering of the genetic correlations to identify five spherical components for the CT and SA ENIGMA summary statistics[9]. As with the current analyses, those analyses indicated that physically adjacent regions tend to cluster together. We build on these findings by characterizing our genomic factors using topographical annotation, Stratified Genomic SEM, and patterns of relationships with external

correlates within the broader SEM framework. GWAS has also been conducted on the first two principal components (PCs) for CT and SA and, in line with our cognitive results, the first principal component for SA was found to be significantly associated with cognitive function[40]. Our findings are conceptually overlapping but statistically distinct from the latter application of PCs in that these are derived based on patterns of phenotypic associations, whereas Genomic SEM explicitly models genetic overlap.

The current analyses have a number of limitations. First, we highlight that our analyses were restricted to participants of European ancestry only due to the availability of sufficiently well-powered GWAS data for this ancestral group coupled with the requirement of LD-score regression to produce estimates within a single ancestral population due to differences in LD structure across groups. It will be of the utmost importance, both with respect to scientific value and representation, that future analyses build on the expanding, genetically informed datasets for different populations. Second, while we were able to evaluate the fit of the factor structure in both ENIGMA and UKB, these were not entirely independent samples. In addition, ENIGMA reflects an age-heterogeneous sample, and findings should be interpreted in this light. These models should continue to be evaluated in external, genetically informed imaging datasets for specific developmental windows. For Stratified Genomic SEM analyses, we utilized the zero-order stratified genetic covariance matrices that do not control for overlap with other annotations for the estimation of enrichment. This decision point reflects the power needed to utilize the $\tau$ matrices that do control for annotation overlap (see "Methods"), but as GWAS sample sizes continue to grow, future work can examine the robustness of these results.

Polygenic risk for schizophrenia was recently found to have more robust associations with microstructural metrics derived from diffusion-weighted imaging (DWI) relative to macrostructural imaging metrics[41]. As our analyses utilized the Desikan–Killiany (DK) atlas parcellations[12], future work could examine both more fine-grained parcellations (e.g., Glasser[42]) and extend analyses to examine phenotypes derived from other scanning modalities, such as DWI-derived outcomes. We note that $F1_{CT}$ mapped onto both physically proximal regions and regions that have previously shown the highest reliability for the DK atlas across manual and automated regional definitions[12]. Future work applying different parcellations may help additionally clarify whether the structure for this particular factor is merely a reflection of a more reliable signal.

Substantially overlapping genetic signal within CT and SA brain regions necessitates statistical tools that allow for examining the multivariate system of relationships across these phenotypes. To this end, we employed Genomic SEM to examine the factor structure and its correlates at the genome-wide level and went on to better characterize these factors by performing topographical annotation and estimating multivariate functional enrichment. In line with prior studies that utilize the global metrics of CT and SA, we find that a general factor explains significant genetic variation and captures patterns of enrichment shared across the 34 brain regions. At the same time, we observe that the five residual genomic factors reflect biologically and functionally relevant partitionings of the cortex. As all data used here is publicly available, we present an accessible framework for studying the multivariate genomic architecture of the cerebral cortex. We propose that future studies use this approach to examine other research questions relevant to this highly studied set of imaging outcomes. Collectively, these findings point towards the utility and need to simultaneously model the different levels of genetic risk sharing, from the most general level across all the cortex to the specific subclusters indexed by the genomic factors, and down to the variation unique to a single brain region.

## Methods

### Quality control procedures

Quality control filters for estimating the genetic covariance and sampling covariance matrices followed the defaults in the Genomic SEM[10] implementation of LDSC[43]. These filters included restricting to SNPs present in HapMap3 and, when this information was available, restricting to minor allele frequency (MAF) >1% and excluding SNPs with information scores (INFO) <0.9. We highlight that UKB and ENIGMA structural summary statistics, and the UKB cognitive summary statistics, included MAF but not INFO. However, ENIGMA filtered summary statistics on imputation quality at the level of the contributing cohorts prior to performing the meta-analysis[9]. Imputation quality was available for a subset of psychiatric disorders. The LD-scores used for LDSC were calculated using the European subsample of the 1000 Genomes phase 3 project; the scores excluded the MHC region due to the high degree of LD outliers which is known to unduly influence estimates. We note also that when calculating the liability scale heritability for the psychiatric traits we used the sum of effective sample sizes, and a sample prevalence of 0.5 to reflect the fact that the corrected sample size already accounts for sample ascertainment; we have shown that this produces a more accurate estimate of heritability for binary traits as it more appropriately accounts for ascertainment differences across cohorts contributing to GWAS meta-analysis[44]. For comparative purposes, population prevalences for the liability conversion were chosen to reflect those used from the corresponding univariate GWAS publication and are reported in Supplementary Data 11.

### Genomic factor analysis: ENIGMA

We refer the reader to the original ENIGMA[9] publication for details about how the univariate GWAS were performed. We note briefly here that ENIGMA[9] used the Desikan–Killiany[12] atlas segmentations to define the 34 brain regions and that SA and CT were measured using T1-weighted magnetic resonance imaging scans. We used the publicly available summary statistics that apply genomic control. We note also that all analyses presented here utilize the GWAS summary statistics that were not corrected for global volume as this allowed for explicitly modeling the shared genetic variation across the 34 regions in the context of a bifactor model. Factor analysis of ENIGMA GWAS summary statistics proceeded in five primary steps. First, standard quality control filters (see "Quality control procedures" section above) were applied to the GWAS summary statistics using the *munge* function in Genomic SEM. Second, LD-score regression[43,45] was applied to the ENIGMA GWAS summary statistics to produce a genetic heatmap across the 34 CT and SA brain regions. As certain SNPs were not present across all cohorts that comprise the ENIGMA consortium, the SNP-specific participant sample sizes were used for LDSC estimation. Third, the Kaiser[15], acceleration factor, and optimal coordinates[16] rules were applied to these genetic correlation matrices in order to collectively determine the number of genomic factors that could be used to parsimoniously represent the data. For both SA and CT, these results pointed towards five factors according to the Kaiser and optimal coordinates tests, and a single, common factor according to the acceleration factor test. Fourth, exploratory factor analyses (EFAs) were conducted using the promax (i.e., correlated factor) rotation in the *factanal* R package.

Finally, we fit confirmatory factor models in Genomic SEM and evaluated these models using standard metrics of model fit[10]. More specifically, comparative fit index (CFI) values above 0.9[17] and standardized root-mean-squared residual (SRMR) values less than 0.10 were considered indicative of acceptable model fit. We also report the Akaike Information Criteria (AIC)[18] a fit index that balances overall model fit with the number of estimated parameters (i.e., parsimony), with lower values indicating better fit. We fit three primary confirmatory models in Genomic SEM. The first was a common factor

model that was used to determine whether, consistent with the acceleration factor test, a single factor was sufficient for describing the data. The second was a five-factor, correlated factors model specified based on the five-factor EFA results. More specifically, individual brain regions were assigned to a factor when their standardized loading was >0.5, or if the brain region did not achieve a loading of 0.5 for any factor assigning the region to the factor with the largest standardized loading. This was with the one exception that the medial orbitofrontal region was the one indicator that showed evidence of cross-loadings for both CT and SA, with standardized loadings >0.5 for both the second and third factor in the CT model and the first and fifth factor in the SA model. However, including this cross-loading in the confirmatory model produced a worse model fit for CT (with cross-loading: AIC = 68157.97, CFI = 0.924, SRMR = 0.070; without cross-loading: AIC = 53570.8, CFI = 0.941, SRMR = 0.070); it also included the only negative factor loadings for CT and caused model convergence issues for SA. This cross-loading was consequently removed for both structural metrics with the medial orbitofrontal factor specified to load on the factor with the highest EFA loading.

The third type of model we fit, and the model presented in the main text, was a bifactor model. This consisted of a general factor that captures shared variation across the 34 brain regions and five, residual factors (defined by the same brain regions from the correlated factors model) that model covariation not accounted for by the general factor. As the general factor defined by all indicators within a bifactor model is conceptually posited to account for the covariation across the remaining factors, the five residual factors were all specified to be orthogonal (i.e., factor correlations fixed to 0). For all models, we used unit variance identification such that the factor variances were fixed to 1. For the SA models, the fourth factor was defined by two brain regions (caudal anterior cingulate and rostral anterior cingulate) and the fifth factor was defined only by the medial orbitofrontal brain region. To ensure that the SA models were locally identified, the factor loadings were then constrained to equality for the fourth factor, and the residual variance of the medial orbitofrontal region that solely defined the fifth factor was fixed to 0.

Given the pervasive levels of genetic overlap across the 34 brain regions, the generally parsimonious representation of the data using five factors, and the stringent threshold of assigning brain regions to factors using standardized loadings of 0.5 or greater, we went on to iteratively add residual covariances across pairs of brain regions. This was done by obtaining the residual covariance matrix—calculated as the difference between the model-implied genetic covariance and observed genetic covariance matrix—and adding the residual covariances one at a time until they no longer reached a significance threshold of $p < 0.01$. This procedure resulted in adding eight residual covariances for CT and seven residual covariances for SA. We confirmed that these residual covariances improved model fit for both the correlated factors and the bifactor model for CT and SA (Supplementary Data 4). We find for both CT and SA that a bifactor model and five-factor correlated factors meet or exceed field standard metrics for providing an acceptable fit to the data and are both considered theoretically informative for downstream analyses, including replication in UK Biobank.

### Genomic factor analysis: UK Biobank

The UK Biobank (UKB; http://www.ukbiobank.ac.uk) is a large population-based cohort study that recruited approximately 500,000 volunteers between 2006 and 2010 across the UK. A subset of participants underwent brain MRI scans since 2014. UKB received ethical approval from the North West Centre Research Ethics Committee (REC number 11/NW/0382). The current analyses were conducted under the approved UKB application 32568.

Raw brain imaging data were processed through an automated image processing pipeline[46] by the UKB imaging team to create a wide

range of image-derived phenotypes (IDPs) (https://www.fmrib.ox.ac.uk/ukbiobank/fbp). Details of the MRI protocol and processing are publicly available[46,47]. Here, we focused on CT and SA measures of the 34 brain regions defined by the Desikan–Killiany atlas[12]. The temporal pole was excluded from analysis, as part of the temporal lobe is difficult to segment, leading to a large amount of missing data for the region.

The genetic data for all UKB participants were subjected to a standard set of QC filters consisting of removing strand ambiguous SNPs, regions with long-range LD, SNPs with call rates <0.98, and SNPs with a minor allele frequency <0.05. Analyses were also restricted to autosomal SNPs. In order to calculate predicted ancestry, PLINK (https://www.cog-genomics.org/plink/2.0/assoc) was first used to perform LD pruning on the QC'd genetic data using an $r^2$ threshold of 0.2 within a 100 kb window that shifted by 50 kb each time. Principal components (PCs) of ancestry calculated in 1000 Genomes Phase 3 data were then projected onto the LD-pruned UKB genetic data. The top six PCs were subsequently used as input to a Random Forest classifier with 1000 Genomes as the training set to calculate predicted probabilities of belonging to a particular ancestral population. Of the initial pool of 40,733 UKB participants with genetically informed MRI data, we retained 31,522 individuals with predicted probabilities of belonging to the European population >90%.

Using sample-level filters created by the original UKB investigators, participant QC was then performed on the European ancestry subsample. This involved removing 3802 individuals with: (1) mismatch between self-reported and genetically inferred sex; (2) missingness or heterozygosity outliers; (3) sex chromosome aneuploidy; or (4) related participants in the sample. Related participants were specifically removed by restricting to participants classified in UKB Data-Field 22020 as having been used for genetic principal component estimation, which was performed on an unrelated sample that removed inferred, third-degree relatives. We note that kinship (i.e., relatedness) was carefully estimated within the full UKB sample using a restricted set of 93,511 SNPs that weakly loaded on principal components of ancestry. This selected set of SNPs is consequently less likely to upwardly bias kinship estimates due to recent admixture[48]. Our approach to filtering relatedness is more conservative than other approaches that attempt to identify the maximum, independent set in a given kinship matrix. However, using the filters provided by UKB investigators is far more computationally efficient as it does not require re-running preprocessing procedures when UKB releases additional neuroimaging data. Finally, we removed 981 individuals with incomplete data on necessary covariates, which yielded a final sample size of 26,739 participants that were brought forward for GWAS analyses (Supplementary Fig. 13 for QC schematic).

GWAS was performed using the non-LD pruned, QC'd genetic and imaging data as input to PLINK. We specifically used the linear regression model, adjusting for age, sex, X/Y/Z/T position of the head and the radio-frequency receive coil in the scanner, UKB imaging acquisition center, mean resting-state and task-based functional MRI head motion, volumetric scaling factor, T1 density, genotyping chip, and the top 40 principal components of the genetic data (estimated within the UKB sample) as covariates.

The ENIGMA imaging sample utilized in the primary analyses also includes an earlier release of the UKB imaging data for $N = 10,083$ participants[9]. While it would be possible to restrict to a strictly independent holdout sample based on the date of the imaging visit in UKB, the ENIGMA paper also included 5095 unrelated European individuals with imaging data from UKB as a holdout sample for polygenic risk score analyses. Consequently, a strict UKB holdout sample based on imaging date would reflect only ~11,500 participants and consequently be underpowered relative to the full UKB imaging sample. As we note in the "Results" section, we view the semi-independent UKB sample as informative both as a replication sample and, more specifically, as a

sample that is far more age homogenous relative to ENIGMA. We also quantify the level of shared information across our UKB summary statistics and the ENIGMA summary statistics using the bivariate LDSC intercept. The bivariate (i.e., cross-trait) LDSC intercept is estimated directly from the GWAS summary data, and for two traits (1 and 2) is expressed as:

$$\frac{\rho_{1,2} N_{s1,2}}{\sqrt{N_1 N_2}} \qquad (1)$$

where $\rho_{1,2}$ is the phenotypic correlation, $N_{s1,2}$ is the sample overlap, and $N_1$ and $N_2$ reflect the total sample size for traits 1 and 2, respectively. The bivariate LDSC intercept then reflects the phenotypic correlation weighed by proportional sample overlap, thereby providing a quantitative index of the sampling dependence across the ENIGMA summary statistics and our UKB summary statistics. We specifically estimated the bivariate intercept for global CT and SA metrics for ENIGMA with the global metrics for the left and right hemispheres in UKB. For CT, the bivariate intercept was 0.178 across ENIGMA and both global metrics of the left and right hemispheres in UKB. For SA, the bivariate intercept was 0.262 across ENIGMA and the left and right global metrics in UKB. As expected, this indicates that our updated UKB summary statistics reflect a largely independent replication cohort relative to the primary ENIGMA analyses.

## Topographical annotation

In order to better understand the spatial organization of our genomic brain factors, we sought to characterize their relationships with established canonical and meta-analytic maps from the neuroimaging literature. The specific test employed was dependent on the nature of the comparison (i.e., comparing two categorical maps required a different test than comparing one categorical map and one continuous map), as described in the following sections. However, regardless of the specific test used in a given comparison, a spin test approach was used to assess the statistical significance of the observed spatial correspondence. We refer the reader to previous publications for a detailed description of this method, but it critically allows for two neuroimaging maps to be compared while accounting for spatial contiguity and hemispheric symmetry of the cortex. Here, we used the spin test approach to generate an empirical null distribution of 10,000 spatially permuted test statistics.

To examine the spatial correspondence between our genomic factors of brain morphology and other categorical neuroimaging maps (e.g., the cytoarchitectural classes defined by von Economo and Koskinas[20,49], the functional intrinsic connectivity networks derived from resting-state functional magnetic resonance imaging [fMRI] by Yeo and Krienen[21,50]), we first assigned labels from each parcellation to corresponding Desikan–Killiany regions on the basis of maximal overlap. We then performed a Fisher's exact test to evaluate the degree of dependence between the two categorical maps of interest. The observed $p$-value from this test was subsequently compared against an empirical null distribution generated via the spin test approach described above, yielding a final $P_{spin}$ value for interpretation.

We also compared the spatial relationships between our genomic factors and numerous features that varied continuously across the cortex, which can be generally described as being derived from either biological/physiological or cognitive/psychological data sources. The former is a collection of cortical maps derived from the BigBrain project (intracortical microstructure, laminar differentiation, cellular/neuronal density)[51–53], a recent meta-analysis of neurotransmitter receptor and transporter densities measured with positron emission tomography[54] and the Allen Human Brain Atlas (cell-type-specific transcriptional signatures)[55], while the latter is a collection of cortical association maps obtained from Neurosynth[56], an online platform for automated meta-analysis of more than 15,000 published fMRI studies.

Here, we used a subset of 123 probabilistic association maps that correspond to cognitive and psychological processes described in the Cognitive Atlas[57] as described in a previous study using similar methods[54]. For these continuous maps, we fit an analysis of variance model and computed an omnibus $F$ statistic for each comparison. As done for the categorical comparisons, $P_{spin}$ values were subsequently calculated and adjusted for false discovery rate.

## Stratified Genomic SEM

Stratified Genomic SEM[11] reflects a multistep process that ends in the ability to estimate enrichment within functional annotations for any parameter of interest (e.g., factor variances) within the genomic model. Functional annotations reflect a subset of genetic variants that are categorized using collateral gene expression data, such as single-cell RNA sequencing, with respect to some shared characteristic, including upward or downward expression within specific tissue types, histone marks, or brain regions. A functional annotation is enriched when the proportion of genetic variation explained by the variants within that annotation is greater than the proportional size of the annotation. The first step in Stratified Genomic SEM is to estimate genetic covariance matrices stratified across a chosen set of annotations. This is achieved by estimating the multivariable version of Stratified LDSC[22] More specifically, stratified genetic covariance within a functional annotation that controls for overlap with other annotations is estimated as:

$$E[z_{1j}z_{2j}] = \sqrt{N_1 N_2} \sum_c \tau_c \frac{\ell(j,c)}{M_c} + \frac{\rho N_s}{\sqrt{N_1 N_2}} + a \qquad (2)$$

where $N_i$ is the sample size for study $i$, $c$ is a specific genomic annotation, $M_c$ is the number of SNPs in the annotation, $\tau_c$ is the coheritability within annotation $c$ controlling for overlap with other annotations, $N_s$ is the sample overlap across the two GWAS studies, $a$ is a constant term across annotations that captures unmeasured confounding (e.g., shared population stratification), and $\rho$ is the phenotypic correlation within overlapping participants. Stratified heritability estimates are produced using the same general formula reduced to the univariate S-LDSC model[15], where the expectation across two $z_j$ statistics for the same trait is given as $E[\chi_j^2]$.

For the current application, the $\tau_c$ values are converted to their zero-order form that does not control for overlap with other annotations. The zero-order stratified values are useful as the estimates are not directly contingent on the other annotations included in the model and produce more stable estimates at moderate GWAS sample sizes, such as those observed here. This comes with the caveat that enriched signal can, in part, reflect signal shared with other annotations and this limitation should be kept in mind when interpreting results. The $\tau_c$ estimates are converted to zero-order values ($\zeta_t$) for a target annotation $t$ by taking the sum of weighted $\tau_c$ values given as:

$$\zeta_t = \sum_c \left( \frac{|M_c \cap M_t|}{|M_c|} \right) \tau_c \qquad (3)$$

where, as in the bivariate S-LDSC equation described above, $|M_c|$ is the total number of SNPs in annotation $c$, and $|M_c \cap M_t|$ is the number of SNPs in both annotations $c$ and $t$. Putting these pieces together, $\left( \frac{|M_c \cap M_t|}{|M_c|} \right) \tau_c$ then reflects the stratified (co)heritability estimate in annotation $c$ weighted by the proportion of annotation $c$ SNPs that are shared with target annotation $t$. The summation of these estimates then produces the zero-order estimates used to populate the zero-order, stratified genetic covariance matrix across the pairwise combinations of included phenotypes.

Each stratified, zero-order genetic covariance matrix is also paired with a stratified, zero-order sampling covariance matrix. The diagonal of the sampling covariance matrix contains squared standard errors of stratified heritability and covariance estimates. The off-diagonals reflect the sampling covariances that capture dependencies among estimation errors that can arise in cases such as participant sample overlap. Both the diagonals and off-diagonals are estimated using a multivariate block jackknife and taken together allow for producing unbiased standard errors in the context of Stratified Genomic SEM. The stratified, zero-order genetic covariance and sampling covariance matrices were specifically estimated using the $s\_ldsc$ function within the $GenomicSEM$ R package.

When the parameter being examined for enrichment reflects pleiotropic effects captured by the factor variances, these stratified matrices are used as input to Genomic SEM wherein the factor loadings are fixed from the model estimated using the genome-wide annotation that includes all SNPs, and the factor and residual indicator variances are freely estimated. The factor variances estimated within an annotation then reflect the proportion of pleiotropic genetic variation captured by a given annotation. This estimate and its standard error are subsequently divided by the proportional size of the annotation to produce estimates of multivariate enrichment.

The current project utilized functional annotations from various sources. This included functional annotations from the most recent 1000 Genomes Phase 3 baseline set of annotations (BaselineLD Version 2.2) recommended by the original developers of univariate S-LDSC[22,58]. These annotations include minor allele frequency bins, histone marks, classes of enhancers and evolutionarily conserved variants, and flanking window annotations. We also include brain and endocrine relevant annotations for tissue-specific gene expression from GTEx[59] and DEPICT[60] and histone marks from the Roadmap Epigenetics project[61]. This was in addition to five randomly selected control regions for gene expression and histone marks (i.e., 10 controls total). Finally, we utilized 29 functional annotations that reflected the main effects of protein-truncating-variant (PTV)-intolerant (PI) genes, human hippocampal and prefrontal brain cells, and their interactions. These annotations were created using collateral data from the Genome Aggregation Database (gnomAD)[62] and GTEx[59] data, with the parameters used to construct these annotations outlined in ref. [11].

Flanking window, control, and continuous annotations were used to produce unbiased estimates of stratified genetic covariance but were excluded from enrichment analyses as these results would not be directly interpretable. This resulted in a total of 168 binary annotations. For SA, we removed 16 annotations that were nonpositive definite and required smoothing the stratified covariance matrix such that any point estimate in the matrix produced a Z-statistic discrepancy >1.96 pre- and post-smoothing. As we examined the enrichment of the genetic variance for six factors (the five residual factors and the general factor from the bifactor model) we employed a Bonferroni correction of $p < 5.48E-5$ (i.e., $0.05_{FDR}/[152_{annotations} \times 6_{Genomic\ Factors}]$). CT required removing 22 annotations due to high degrees of matrix smoothing, but we employed the same Bonferroni correction of $p < 5.48E-5$ for comparative purposes.

## Genetic overlap with cognitive function

Genetic overlap with genetic $g$ was examined in the context of the finalized bifactor and correlated factors models (i.e., those that included residual covariance across specific brain regions) for both CT and SA. This was achieved by simultaneously estimating all brain factors with $g$-factor correlations in the context of the models. As a first step in these analyses, we estimated an omnibus model $\chi^2$ difference test that compared the model fit for a model in which all correlations with the brain-based factors were freely estimated versus constrained to equality. This omnibus test asks, at a general level, whether $g$ shows a uniform or factor-specific pattern of relationships for CT or SA. In line with the model fitting procedure used for the brain-regions-only models, residual covariances between individual brain regions and cognitive tests were iteratively added until they no longer reached a

significance threshold of $p < 0.01$ within the context of the correlated factors model. For CT, this resulted in adding two residual covariances between verbal numerical reasoning and both the insula and pre-central region. No residual covariances were added for SA. As each family of tests for CT and SA with $g$ consisted of 204 possible associations (i.e., 34 brain regions × 7 cognitive tests) we employed a Bonferroni correction of 0.05/204 ($p < 2.10E-4$).

## Genetic overlap with psychiatric disorders

The same four-factor, correlated factors model from ref. [11] was used to model the multivariate architecture across the same 11 disorders. We note a few differences in the summary statistics used for the 11 disorders paper relative to the current analyses. This includes utilizing only publicly available summary statistics such that 23andMe data was not included for ADHD or MDD, using the most recent Freeze 3 summary statistics for bipolar disorder[28], and using the recently released anxiety summary statistics that reflect meta-analysis across the UK Biobank, ANGST, and iPSYCH consortium[32]. We began by confirming that the model for psychiatric disorders only still provided a good fit to the data even with the noted differences for some summary statistics (AIC = 162.86, CFI = 0.976, SRMR = 0.097), and that the factor loadings were largely concordant with those reported in prior work (Supplementary Data 10).

As with the $g$-factor analyses, we went on to examine the factor correlations across all psychiatric factors and added residual covariance between individual disorders and brain regions until they no longer reached a significance threshold of $p < 0.01$. For SA, this resulted in adding in four residual relationships between autism spectrum disorder and the temporal pole, pars orbitalis, transverse temporal, and rostral anterior cingulate regions. No residual covariances were identified for CT. As each family of tests for CT and SA with psychiatric disorders consisted of 374 possible associations (i.e., 34 brain regions × 11 disorders), we employed a Bonferroni correction of 0.05/374 ($p < 1.34E-4$). We also estimated the same omnibus model $\chi^2$ difference test by comparing a model in which the correlations across the brain-based and psychiatric factors were fixed to equality versus freely estimates.

### Reporting summary

Further information on research design is available in the Nature Portfolio Reporting Summary linked to this article.

## Data availability

The data that support the findings of this study are all publicly available or can be requested for access. Specific download links for various datasets are directly below. Summary statistics for ENIGMA are available from: http://enigma.ini.usc.edu/research/download-enigma-gwas-results/. Summary statistics for the seven, individual cognitive traits are available from: https://datashare.is.ed.ac.uk/handle/10283/3756. Summary statistics for data from the PGC can be downloaded or requested here: https://www.med.unc.edu/pgc/download-results/. Summary statistics for the Anxiety phenotype can be downloaded here: https://drive.google.com/drive/folders/1fguHvz7l2G45sbMI9h_veQun4aXNTy1v. Data from gnomAD used to identify PI genes for the creation of annotations can be downloaded here: https://console.cloud.google.com/storage/browser/_details/gcp-public-data–gnomad/release/2.1.1/constraint/gnomad.v2.1.1.lof_metrics.by_gene.txt.bgz?pageState=(%22StorageObjectListTable%22:(%2ff%22:%22%255B%255D%22)). Gene count data per cell for the creation of annotations were obtained from: https://storage.googleapis.com/gtex_additional_datasets/single_cell_data/GTEx_droncseq_hip_pcf.tar. Data that maps individual cells to cell types (e.g., neuron, astrocyte etc.) were obtained from: https://static-content.springer.com/esm/art%3A10.1038%2Fnmeth.4407/MediaObjects/41592_2017_BFnmeth4407_MOESM10_ESM.xlsx. Links to the LD-scores, reference panel data, and the code used to

produce the current results can all be found at: https://github.com/GenomicSEM/GenomicSEM/wiki. Links to the BaselineLD v2.2 annotations can be found here: https://data.broadinstitute.org/alkesgroup/LDSCORE/. Data for the UK Biobank (UKB) can be requested from: https://bbams.ndph.ox.ac.uk/ams/signup. Cortical maps of intracortical microstructure, laminar differentiation, and cellular/neuronal density were generated using data from the Big-Brain project at: https://bigbrainproject.org/maps-and-models.html. Cortical maps of neurotransmitter receptor and transporter densities can be obtained from: https://github.com/netneurolab/hansen_receptors. Probabilistic association cortical maps of 123 cognitive and psychological processes (generated using data from Neursynth [https://neurosynth.org/]) can be obtained from: https://github.com/netneurolab/hansen_receptors. Cortical maps of cell-type-specific transcriptional signatures were generated using data from the Allen Human Brain Atlas at: https://human.brain-map.org/static/download.

## Code availability

GenomicSEM software (which now includes the *write.model* functionality to generate model syntax based on EFA output for extensive sets of traits), is an R package that is available from GitHub at the following URL: https://github.com/GenomicSEM/GenomicSEM. Directions for installing and using the GenomicSEM R package can be found at: https://github.com/GenomicSEM/GenomicSEM/wiki. The specific Genomic SEM code release used for these analyses can be found here: https://doi.org/10.5281/zenodo.7512951. The PLINK 2.0 software used to run GWAS analyses in UK Biobank can be downloaded here: https://www.cog-genomics.org/plink/2.0/. The factanal software used to fun the EFAs can be downloaded directly in R, with its documentation provided here: https://www.rdocumentation.org/packages/stats/versions/3.6.2/topics/factanal. The figures that overlay the current findings over images of the human brain (e.g., Fig. 1c) were created using the BrainsForPublication software: https://github.com/WhitakerLab/BrainsForPublication.

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

## Acknowledgements

The work presented here was supported by a gift from the Tommy Fuss Fund. A.D.G. was supported by NIH Grants R01MH120219 and RF1AG073593. We add that the current analyses would not have been possible without the enormous efforts put forth by the investigators and participants from ENIGMA, the Psychiatric Genetics Consortium, iPSYCH, and UK Biobank.

## Author contributions

Study design: A.D.G., T.T.M., Z.L., J.S., T.G., and J.W.S. Genome-wide association study in UKB: Z.L. and T.G. Genetic factor modeling: A.D.G. and T.T.M. Multivariate enrichment analyses: A.D.G. Topographical annotation: T.T.M., J.S., and Z.L. Writing: A.D.G., T.T.M., Z.L., J.S., T.G., and J.W.S.

## Competing interests

J.W.S. is a member of the Leon Levy Foundation Neuroscience Advisory Board, the Scientific Advisory Board of Sensorium Therapeutics, and has received honoraria for internal seminars at Biogen, Inc and Tempus Labs. He is PI of a collaborative study of the genetics of depression and bipolar disorder sponsored by 23andMe for which 23andMe provides analysis time as in-kind support but no payments. The remaining authors declare no competing interests.
