## [Peer Review File · Nature Communications]

Multivariate Genomic Architecture of Cortical Thickness and Surface Area at Multiple Levels of AnalysisEditorial Note: This manuscript has been previously reviewed at another journal that is not operating a transparent peer review scheme. This document only contains reviewer comments and rebuttal letters for versions considered at *Nature Communications* .

REVIEWER COMMENTS

Reviewer #2 (Remarks to the Author):

We thank the authors for their responses to the reviewer's comments. The manuscript has improved and is clearer now.

Cortical volume was mentioned a number of times by the authors and R1. The authors may wish to consider that it was deliberate decision not to cortical ROI volumes in the ENIGMA paper, this decision was driven by 1) the very high correlation between Cortical SA and Vol and 2) the fact that anyone wishing to obtain a GWAS of cortical volumes was expected to do this for themselves using one of the multivariate secondary analysis packages (at the time we were expecting people to use GWIS).

Given that the take home of the paper is that there are five main factors I find the suggestion that finer grained atlases should be used rather odd. Can the authors explain why/how they would expect the results to change if a different atlas is used? I note R1 also comments "this is evidence that increased precision may not improve associations?"

UKB QC – this is still an extremely large number of participants to drop for these QC steps (and many more than I or the UKB themselves drop applying the same QC steps) can the authors please specify how many people are being dropped at each step and confirm that the steps are being undertaken sequentially (there are known problems of trying to drop related individuals if multiple ancestries are present in a sample).

Independent UKB sample – we thank the authors for considering this point and moderating their language.

Reviewer #3 (Remarks to the Author):

The authors have addressed all my previous concerns. The manuscript is well written. The authors performed a wide range of analyses carefully.

A minor point for data interpretation -

The authors report null results for genetic overlap between brain morphology and psychiatric disorders and use ref 36 to support interpretation. "These null results are in-line with recent findings indicating that the majority of associations across structural metrics and human complex traits are much smaller than initially thought and that the bulk of prior studies have been underpowered.³⁶". However, data interpretation can benefit from other recent relevant literature. For example, ENIGMA meta-studies show robust associations between brain morphology and psychiatric disorders, though at the phenotypic level (PMID: 32198361). It could be that this current study is still underpowered to detect genetic overlap. The authors state that this study is well-powered, which seems to be overstated.

Reviewer #2

We thank the authors for their responses to the reviewer's comments. The manuscript has improved and is clearer now.

Cortical volume was mentioned a number of times by the authors and R1. The authors may wish to consider that it was deliberate decision not to cortical ROI volumes in the ENIGMA paper, this decision was driven by 1) the very high correlation between Cortical SA and Vol and 2) the fact that anyone wishing to obtain a GWAS of cortical volumes was expected to do this for themselves using one of the multivariate secondary analysis packages (at the time we were expecting people to use GWIS). Given that the take home of the paper is that there are five main factors I find the suggestion that finer grained atlases should be used rather odd. Can the authors explain why/how they would expect the results to change if a different atlas is used? I note R1 also comments "this is evidence that increased precision may not improve associations?"

We thank the reviewer for these suggestions. After considering these points we have provided a reference for why more fine-grained parcellations might be expected to produce different results. In addition, we replace the mention of cortical volume to discuss DWI derived phenotypes as an exemplar, alternative structural metric that could be examined in future analyses. We now write on p.9:

"Polygenic risk for schizophrenia was recently found to have more robust associations with micro-structural metrics derived from diffusion-weighted imaging (DWI) relative to macro-structural imaging metrics.⁴¹ As our analyses utilized the Desikan-Killiany (DK) atlas parcellations,¹² future work could examine both more fine-grained parcellations (e.g., Glasser⁴⁰) and extend analyses to examine other forms of structural metrics, such as DWI derived phenotypes."

UKB QC – this is still an extremely large number of participants to drop for these QC steps (and many more than I or the UKB themselves drop applying the same QC steps) can the authors please specify how many people are being dropped at each step and confirm that the steps are being undertaken sequentially (there are known problems of trying to drop related individuals if multiple ancestries are present in a sample).

We agree that greater detail was needed and have updated the Method section to provide further details about the QC of the genetic and participant data within UKB and the order in which those QC filters were applied. We now write on page 13:

"The genetic data for all UKB participants were subjected to a standard set of QC filters consisting of removing strand ambiguous SNPs, regions with long-range LD, SNPs with call rates < 0.98, and SNPs with a minor allele frequency < .05. Analyses were also restricted to autosomal SNPs. In order to calculate predicted ancestry, PLINK (<https://www.cog-genomics.org/plink/2.0/assoc>) was first used to perform LD pruning on the QC'd genetic data using an r^2 threshold of 0.2 within a 100kb window that shifted by 50kb each time. Principal components (PCs) of ancestry calculated in 1000 Genomes Phase 3 data were then projected onto the LD-pruned UKB genetic data. The top 6 PCs were subsequently used as input to a Random Forest classifier with 1000 Genomes as the training set to calculate predicted probabilities of belonging to a particular ancestral population. Of the initial pool of 40,733 UKB participants with genetically informed MRI data, we retained 31,522 individuals with predicted probabilities of belonging to the European population > 90%.

Using sample-level filters created by the original UKB investigators, participant QC was then performed on the European ancestry subsample. This involved removing 3,802 individuals with: (i) mismatch between self-reported and genetically inferred sex; (ii) missingness or heterozygosity outliers; (iii) sex chromosome

aneuploidy; or (iv) related participants in the sample. Related participants were specifically removed by restricting to participants classified in UKB Data-Field 22020 as having been used for genetic principal component estimation, which was performed on an unrelated sample that removed inferred, 3rd degree relatives. We note that kinship (i.e., relatedness) was carefully estimated within the full UKB sample using a restricted set of 93,511 SNPs that weakly loaded on principal components of ancestry. This selected set of SNPs is consequently less likely to upwardly bias kinship estimates due to recent admixture.⁴⁸ Our approach to filtering relatedness is more conservative than other approaches that attempt to identify the maximum, independent set in a given kinship matrix. However, using the filters provided by UKB investigators is far more computationally efficient as it does not require re-running preprocessing procedures when UKB releases additional neuroimaging data. Finally, we removed 981 individuals with incomplete data on necessary covariates, which yielded a final sample size of 26,739 participants that were brought forward for GWAS analyses (Supplementary Figure 13 for QC schematic).

GWAS was performed using the non-LD pruned, QC'd genetic and imaging data as input to PLINK. We specifically used the linear regression model, adjusting for age, sex, X/Y/Z/T position of head and the radio-frequency receive coil in the scanner, UKB imaging acquisition center, mean resting-state and task-based functional MRI head motion, volumetric scaling factor, T1 density, genotyping chip and the top 40 principal components of the genetic data (estimated within the UKB sample) as covariates.”

Independent UKB sample – we thank the authors for considering this point and moderating their language.

Of course!

Reviewer #3:

The authors have addressed all my previous concerns. The manuscript is well written. The authors performed a wide range of analyses carefully.

A minor point for data interpretation: The authors report null results for genetic overlap between brain morphology and psychiatric disorders and use ref 36 to support interpretation. “These null results are in-line with recent findings indicating that the majority of associations across structural metrics and human complex traits are much smaller than initially thought and that the bulk of prior studies have been underpowered.³⁶”. However, data interpretation can benefit from other recent relevant literature. For example, ENIGMA meta-studies show robust associations between brain morphology and psychiatric disorders, though at the phenotypic level (PMID: 32198361). It could be that this current study is still underpowered to detect genetic overlap. The authors state that this study is well-powered, which seems to be overstated.

We thank the reviewer for pointing this out. We have updated that section of the Discussion to consider the particular study highlighted and consider the possibility that these findings are still underpowered (p. 9):

“Conversely, we observe no significant genetic relationships across CT or SA and the four psychiatric factors. These null results are in-line with recent findings indicating that the majority of associations across structural metrics and human complex traits are much smaller than initially thought and that the bulk of prior studies have been underpowered.³⁶ At the same time, large-scale phenotypic meta-analyses indicate widespread associations across structural metrics and various psychiatric disorders.³⁷ These findings may reflect associations that operate through largely environmental pathways or the current analyses may not be

sufficiently powered to detect genetic effects for psychiatric disorders. It is also possible that psychiatric-structural associations are specific to different parcellations of the cortex or to clinically ascertained samples. In line with this latter account, associations between CT and various psychiatric disorders have been shown to reflect responses to treatment.^{38,39}”

REVIEWERS' COMMENTS

Reviewer #2 (Remarks to the Author):

The authors have addressed my comments well.

One last comment (which you can choose to ignore). As T1 and DWI are different scanning protocols you may want to rephrase this sentence

As our analyses utilized the Desikan-Killiany (DK) atlas parcellations,¹² future work could examine both more fine grained parcellations (e.g., Glasser⁴⁰) and extend analyses to examine other forms of structural metrics, such as DWI derived phenotypes.”

to be something along the lines of

As our analyses utilized the Desikan-Killiany (DK) atlas parcellations,¹² future work could examine both more fine grained parcellations (e.g., Glasser⁴⁰) and extend analyses to examine phenotypes derived from other scanning modalities such as DWI.”

Reviewer #2 (Remarks to the Author):

The authors have addressed my comments well.

One last comment (which you can choose to ignore). As T1 and DWI are different scanning protocols you may want to rephrase this sentence

As our analyses utilized the Desikan-Killiany (DK) atlas parcellations,¹² future work could examine both more fine grained parcellations (e.g., Glasser⁴⁰) and extend analyses to examine other forms of structural metrics, such as DWI derived phenotypes.”

to be something along the lines of

As our analyses utilized the Desikan-Killiany (DK) atlas parcellations,¹² future work could examine both more fine grained parcellations (e.g., Glasser⁴⁰) and extend analyses to examine phenotypes derived from other scanning modalities such as DWI.”

We thank the reviewer for this suggestion and agree that the proposed wording is more accurate. The corresponding text has been updated in the Discussion section on p. 9:

“As our analyses utilized the Desikan-Killiany (DK) atlas parcellations,¹² future work could examine both more fine-grained parcellations (e.g., Glasser⁴²) and extend analyses to examine phenotypes derived from other scanning modalities, such as DWI derived outcomes.”